# Locoregional Treatments for Metastatic Gastrointestinal Stromal Tumor in British Columbia: A Retrospective Cohort Study from January 2008 to December 2017

**DOI:** 10.3390/cancers14061477

**Published:** 2022-03-14

**Authors:** Tiffany Patterson, Haocheng Li, Jocelyn Chai, Angeline Debruyns, Christine Simmons, Jason Hart, Phil Pollock, Caroline L. Holloway, Pauline T. Truong, Xiaolan Feng

**Affiliations:** 1Clinical Trials, BC Cancer—Vancouver Island Center, Victoria, BC V8R 6V5, Canada; tiffany.patterson@bccancer.bc.ca (T.P.); philip.pollock@bccancer.bc.ca (P.P.); 2Department of Mathematics and Statistics, University of Calgary, Calgary, AB T2N 1N4, Canada; haocheng.li@ucalgary.ca; 3Department of Medicine, University of British Columbia, Vancouver, BC V1Y 1T3, Canada; jocelynchai@alumni.ubc.ca; 4Department of Medicine, Island Medical Program, University of British Columbia, Victoria, BC V1Y 1T3, Canada; angeline.debruyns@gmail.com; 5Department of Medical Oncology, University of British Columbia, BC Cancer—Vancouver Center, Vancouver, BC V1Y 1T3, Canada; christine.simmons@bccancer.bc.ca; 6Department of Medical Oncology, University of British Columbia, BC Cancer—Vancouver Island Center, Victoria, BC V1Y 1T3, Canada; jhart@bccancer.bc.ca; 7Department of Radiation Oncology, University of British Columbia, BC Cancer—Vancouver Island Center, Victoria, BC V1Y 1T3, Canada; cholloway@bccancer.bc.ca (C.L.H.); ptruong@bccancer.bc.ca (P.T.T.); 8Department of Medical Oncology, Tom Baker Cancer Center, Calgary, AB T2N 4N2, Canada; 9Cumming School of Medicine, University of Calgary, Calgary, AB T2N 4N1, Canada

**Keywords:** gastrointestinal stromal tumor (GIST), localregional treatment (LRT), surgery, radiation treatment (RT), local ablation, Tyrosine Kinase Inhibitor (TKI)

## Abstract

**Simple Summary:**

It is not known if surgery, radiation treatment (RT) or other types of locolregional treatment (LRT) may be beneficial for patients with metastatic gastrointestinal stromal tumor (mGIST) in addition to systemic treatment. Our study aims to address this question by analyzing a cohort of 127 mGIST patients in British Columbia over a decade (from January 2008 to December 2017). We showed that mGIST patients who underwent surgery and LRT seemed to have better survival when compared to patients who did not undergo surgery and LRT. However, this treatment strategy should only be considered in patients with limited volume metastatic disease or oligoprogression while the rest of the disease is well controlled with systemic treatment. In addition, RT can offer palliative benefits such as pain relief and bleeding control. Our study, consistent with other retrospective studies, supports LRT consideration in selected mGIST patients within a multidisciplinary setting. This approach is not considered as a “standard of care” due to lack of prospective clinical trials but may improve clinical outcome for some mGIST patients.

**Abstract:**

Introduction: The role of surgery and non-surgical locoregional treatments (LRT) such as radiation therapy (RT) and local ablation techniques in patients with metastatic gastrointestinal stromal tumor (GIST) is unclear. This study examines LRT practice patterns in metastatic GIST and their clinical outcomes in British Columbia (BC). Methods: Patients diagnosed with either recurrent or de novo metastatic GIST from January 2008 to December 2017 were identified. Clinical characteristics and outcomes were analyzed in patients who underwent LRT, including surgical resection of the primary tumor or metastectomy, RT, or other local ablative procedures. Results: 127 patients were identified: 52 (41%) had de novo metastasis and 75 (59%) had recurrent metastasis. Median age was 67 (23–90 years), 58.2% were male, primary site was 33.1% stomach, 40.2% small intestine, 11% rectum/pelvis, and 15.7% others. 37 (29.1%) of patients received palliative surgery, the majority of which had either primary tumor removal only (43.3%) or both primary tumor removal and metastectomy (35.1%). A minority of patients underwent metastectomy only (21.6%). A total of 12 (9.5%) patients received palliative RT to metastatic sites only (58.3%) or primary tumors only (41.7%), mostly for symptomatic control (*n* = 9). A few patients (*n* = 3) received local ablation for liver metastatic deposits with 1 patient receiving microwave ablation (MWA) and 2 receiving radiofrequency ablation (RFA). Most patients (*n* = 120, 94.5%) received some type of systemic treatment. It is notable that prolonged progression free survival (PFS) was observed for the majority of patients who underwent surgery in the metastatic setting with a median PFS of 20.5 (95% confidence interval (CI): 14.29–40.74) months. In addition, significantly higher median overall survival (mOS) was observed in patients who underwent surgery (97.15 months; 95% CI: 77.7-not reached) and LRT (78.98 months; 95% CI: 65.58-not reached) versus no surgery (45.37 months; 95% CI: 38.7–64.69) and no LRT (45.27 months; 95% CI: 33.25–58.66). Almost all patients (8 out of 9) achieved symptomatic improvement after palliative RT. All 3 patients achieved partial response and 2 out of 3 patients had relatively durable responses of 1 year or more after local ablation. Discussion: This study is among the first to systematically examine the use of various LRT in metastatic GIST management. Integration of LRT with systemic treatments may potentially provide promising durable response and prolonged survival for highly selected metastatic GIST patients with low volume disease, limited progression and otherwise well controlled on systemic treatments. These observations, consistent with others, add to the growing evidence that supports the judicious use of LRT in combination with systemic treatments to further optimize the care of metastatic GIST patients.

## 1. Introduction

The treatment of metastatic gastrointestinal stromal tumor (GIST) has significantly evolved since the introduction of imatinib to the therapeutic realm in 2002 [1,2]. GIST is considered a paradigm for precision oncology because imatinib, which specifically targets KIT/PDGFR driver mutations, has drastically improved the median overall survival (mOS) to around 5 years as well as the quality of life of patients diagnosed with metastatic GIST, which was once recognized as a devastating chemotherapy-resistant cancer with a dismal prognosis of less than 6 months [3]. In addition to imatinib, sunitinib and regorafenib are considered standard second- and third-line treatments, respectively for metastatic GIST [4,5]. Recently, Ripretinib was approved by the US Food and Drug Administration (FDA) as a fourth-line treatment [6]. Furthermore, avapritinib was approved by the FDA as a first line and beyond treatment for metastatic GIST harboring the PDGFR D842 mutation [7]. A number of other investigational drugs including carbozatinib are on the horizon and are actively tested in clinical trials [8,9].

Most metastatic GIST will eventually develop resistance to systemic treatments, a common clinical problem, likely due to inherent or acquired secondary mutations in the molecular driver of GIST [10]. However, a subset of metastatic patients demonstrate a unique pattern of slow and limited disease progression, where progression only occurs at one or a few tumor sites (i.e., oligoprogression) over a period of time while the majority of other sites of disease remain controlled by systemic treatments. In this context, locoregional treatments (LRT) such as surgery, radiotherapy (RT), and other interventional radiological ablative procedures may play a role in eradicating progressive disease caused by resistant tumor clones while enabling the continuation of systemic treatment to control stable disease (SD) in sensitive tumor clones. This approach has been widely considered and adopted across many cancer types including GIST despite a lack of prospective randomized phase III clinical trials, which is considered a “gold standard” to approve a therapeutic approach [11,12,13,14,15,16].

The current study aims to evaluate practice patterns of surgery, RT and other local ablation techniques such as radiofrequency ablation (RFA) and microwave ablation (MWA) in metastatic GIST patients diagnosed between January 2008 to December 2017 in British Columbia (BC).

## 2. Methodology

### 2.1. Study Subjects

All study subjects were patients referred to BC Cancer from January 2008 to December 2017 with newly diagnosed metastatic GIST, identified using the BC Cancer Registry and BC Cancer Sarcoma Outcome Unit.

### 2.2. Data Collection and Analysis

Chart review was performed to extract data on demographics, date of pathological diagnosis, tumor characteristics, treatment modalities, date of recurrence if curative surgery was performed, date of response and progression, date of death, and date of last follow-up. Treatment response was determined using the Response Evaluation Criteria in Solid Tumors (RECIST; version 1.1) [17].

### 2.3. Statistical Analysis

Descriptive analysis was performed to characterize the demographics, tumor characteristics and treatment of the cohort. Progression-free survival (PFS) and OS outcomes for patients with metastatic disease were estimated using the Kaplan–Meier method. Subgroup comparisons were performed using log–rank tests. Cox proportional hazards regression methods were used to compare OS between groups in a multivariate model incorporating clinical prognostic factors such as age, sex, tumor size, mitotic count and metastatic status (de novo versus recurrent). All statistical analyses were performed by software R v3.3.0 (R Foundation for Statistical Computing, Vienna, Austria) [18].

### 2.4. Ethics

The study was approved by the University of British Columbia Research Ethics Board (REB# H19-02339).

## 3. Results

### 3.1. Cohort Characteristics

127 metastatic GIST patients were identified with a median follow up of 79.6 months (range: 68.5–122.2 months). A total of 41% (*n* = 52) had de novo metastatic disease and 59% (*n* = 75) had recurrent metastatic disease. The median age at metastatic diagnosis was 67 years (range 23 to 90 years), and 58.2% (*n* = 74) were males. Most patients were treated at BC Cancer, with 1 patient treated in the community (Table 1).

Among patients with recurrent metastatic disease, most presented with primary tumor size greater than 10 cm (*n* = 37, 49.3%), mitotic rate greater than 10/50 HPF (*n* = 33, 44.0%), tumor location in small intestine (*n* = 32, 42.7%), and a fraction of patients also experienced a tumor rupture (*n* = 7, 9.3%). No significant differences in primary tumor characteristics were observed between patients with and without LRT. Mutational analysis (MA) was similarly performed for the majority of patients with (80%) and without LRT (74.5%). Furthermore, no statistically significant difference of mutational types was observed among these two groups (LRT versus no LRT) (Table 1, data not shown).

For patients with de novo metastatic disease, similar to the recurrent metastatic cohort, most presented with primary tumor with high-risk features including tumor size greater than 10 cm (*n* = 22, 42.3%) or between 5–10 cm (*n* = 22, 42.3%), mitotic rate greater than 10/50HPF (*n* = 17, 32.7%), tumor primarily located in the small bowel (*n* = 19, 36.5%), and four patients also experienced a tumor rupture (*n* = 4, 7.7%). In addition, similar to the recurrent metastatic cohort, no statistically significant difference was observed in terms of primary tumor characteristics between de novo metastatic patients with and without LRT (Table 1, data not shown). Interestingly, a lower percentage of MA use was noted in de novo metastatic patients with LRT (65.4%) and without LRT (57.7%) when compared to the recurrent metastatic cohort (see above). No statistically significant difference of mutational types were observed between these two groups (LRT versus no LRT) (Table 1, data not shown).

The majority of patients (*n* = 120, 94.5%) received first line treatment, mostly with imatinib (*n* = 112, 93.3%), 59 (46.4%) patients received second line treatment, mostly with sunitinib (*n* = 52, 88.1%), 33 (26%) patients received third line treatment mostly with regorafenib (*n* = 22, 66.7%), and 15 (11.8%) patients received fourth line treatment (*n* = 15) including imatinib rechallenge (*n* = 8, 53.3%), cabozantinib (*n* = 3, 20%), regorafenib (*n* = 2, 13.3%), sorafenib (*n* = 1, 6.7%), and nilotinib (*n* = 1, 6.7%). For the few patients that received fifth line treatment (*n* = 4, 3.1%), each received a different drug including avapritinib (*n* = 1, 25%), imatinib (*n* = 1, 25%), nilotinib (*n* = 1, 25%), and ripretinib (*n* = 1, 25%). Only one patient with de novo metastatic disease without LRT received sixth line treatment which consisted of imatinib rechallenge (Appendix A). No statistically significant difference was observed between recurrent versus de novo metastatic patients with or without LRT in terms of drug of choice and sequence of systemic treatment (Table 1 and data not shown), however, most metastatic patients who underwent LRT were on first line imatinib treatment which will be discussed later.

### 3.2. Surgery

It is notable that a sizable portion of metastatic patients in our cohort received palliative surgery (*n* = 37, 29.1%) (Table 1). The majority of these were either primary tumor removal only (*n* = 16, 43.3%) or both primary tumor removal and metastectomy (*n* = 13, 35.1%). A minority of patients underwent metastectomy only (*n* = 8, 21.6%). A total of four patients underwent primary tumor resection in emergency (i.e., acute bleeding) or palliative situations (i.e., obstruction/pain control). They all had acute symptom improvements after palliative surgery. However, these 4 patients were excluded from the survival analysis. In all other cases, primary tumor removal in the metastatic setting was offered to patients with good performance status, and a small volume of metastatic disease which was well controlled on systemic treatment. Overall, median PFS (mPFS) of patients who underwent surgery in our cohort was 20.5 months (95% confidence interval (CI):14.29 to 40.74 months) (Figure 1A). We further examined PFS specifically in various surgical situations. Most patients that had only primary tumor removal either remained with no evidence of disease (NED) or durable SD of more than 3 years as of their last follow up (*n* = 9) or had prolonged PFS of between 1 to 3 years. The other four patients had shorter PFS between 7–9 months and had either incomplete resections of their primary tumor and/or significant metastatic disease left in situ. Metastectomy was performed in our cohort (*n* = 8) primarily with “curative intent” to remove all residual disease to achieve NED while their primary tumor was removed either in the previous curative setting or current metastatic setting. In fewer cases (*n* = 4), metastectomies were performed due to oligo disease progression with systemic treatment while the remaining other metastatic disease was controlled well with the goal of eradicating resistant clones to prolong effective TKI treatment. Of the 13 patients that had both primary tumor and metastatic disease removed, three patients remain NED, one patient with durable SD of more than 6 years as of last follow up, six patients with PFS of 1 year or more, while only two patients with a relative shorter PFS of 3–6 months, and one patient died of unrelated causes. Of the seven patients that only had metastasectomies, one patient remained with NED, three patients had PFS of more than 1 year, while three patients had a PFS of 6–12 months who all had significant residual metastatic disease after surgery. Seventeen patients (45.9%) who underwent surgery in the metastatic setting were on first line imatinib (*n* = 13, 76.5%) prior to surgery and continued on imatinib after surgery. Only a minority of patients (*n* = 4, 23.5%) were on sunitinib. Systemic treatment was stopped for 1–2 weeks during surgery.

### 3.3. Radiotherapy (RT)

In our cohort of patients who received palliative RT (*n* = 12), it was primarily given at metastatic sites (*n* = 7) for pain (*n* = 2), spinal cord compression (*n* = 2), hemoptysis (*n* = 1) and oligo disease progression (*n* = 2). A total of three patients experienced complete symptomatic resolution (*n* = 2 pain resolved, *n* = 1 no hemoptysis), while two other patients with impending cord compression benefited with symptomatic improvement for 3 and 6 months, respectively (*n* = 2), However, the patients that received RT for oligo disease progression (*n* = 2) either progressed during RT treatment or in the irradiated area 1 month later. Only one patient achieved local tumor control with SD for 6 months (Table 2).

Palliative RT that was administered to primary tumors (*n* = 5) was mainly given for bleeding (*n* = 3) and pain (*n* = 2). Patients that received RT for bleeding (*n* = 3) all had symptom improvement. Of those three patients, one received RT for preventative bleeding as the patient was on anticoagulation medication for treatment of splenic vein thrombus. One patient who was irradiated for pain control did not have symptom improvement while the other experienced symptom improvement.

### 3.4. Local Ablation

In our cohort, local ablation (RFA = 2, MWA = 1) was given for liver metastatic deposits for oligo disease progression (*n* = 2) and eradication of residual disease (*n* = 1). The patient that received MWA had PFS of 17 months; one patient that received RFA to eradicate residual disease had PFS of 12 months, while another patient that received RFA for oligo disease progression had PFS of only 1 month but had bulkier hepatic and extrahepatic disease than the other two patients. All three patients continued imatinib during local ablative treatments. No safety concerns were observed for these three patients (Table 3).

### 3.5. Survival Outcome

Four metastatic patients were excluded from the survival analysis of this study because they underwent emergency surgery due to either acute bleeding, perforation, obstruction or pain control. It is notable that metastatic patients (*n* = 33) who underwent surgery electively had a mOS of 97.15 (95% CI: 77.7-not reached) months, which is more than doubled when compared to a mOS 45.37 (95% CI: 38.7–64.69) months for metastatic patients who did not undergo surgery (Figure 2A). Similar differences were observed in both recurrent and de novo metastatic patients (Figure 2B,C). Similar mOS differences are observed between metastatic patients who underwent LRT (*n* = 42) versus not (*n* = 81) with a mOS of 86.37 (95% CI: 75.14-not reached) months and a mOS 45.27 (95% CI: 33.25–58.66) months respectively (Figure 3A). Again, similar findings were observed in both recurrent and de novo metastatic patients (Figure 3B,C). In addition, multivariate Cox regression analysis revealed the hazard ratio (HR) of mOS between metastatic patients who underwent surgery versus not was 0.343 (*p* = 0.0004) without adjustment, HR remained significant at 0.393 (*p* = 0.004) after adjustment of age, sex, tumor size, mitotic count as well as metastatic status (de novo versus recurrent). Similarly, HR of mOS between metastatic patients who underwent LRT versus not was 0.416 (*p* = 0.0011) without adjustment and HR remained significant at 0.46 (*p* = 0.01) after adjustment.

## 4. Discussions

### 4.1. Surgery

The role of surgery in metastatic sarcoma has been established and is frequently used in practice. However, it is not considered a “standard of care” in metastatic GIST management. Currently, large prospective randomized controlled trials (RCT) in the literature studying the long-term outcomes of either primary tumor removal in the metastatic setting or metastectomy and its additional benefit on patients already on TKI therapy is lacking. One small RCT from China with a total of 41 patients enrolled (211 patients planned) had to close early due to poor accrual but reported numerically better survival outcomes in patients who underwent surgery to remove macroscopic disease as completely as possible compared to those who did not (2 years PFS of 88.4% versus 57.7% respectively, *p* = 0.024) [19]. Another RCT was attempted in Europe but was unable to recruit enough participants, with no published data to date (NCT00956072 clinicaltrials.org). There are, however, several retrospective studies that appear to consistently support the role of surgery in selected metastatic GIST patients. One study reported significant long term survival outcomes for patients who underwent near complete resection (R0/R1) compared to those did not (R2) (mOS was 8.7 years for R0/R1 and 5.3 y in patients with R2 resection (*p* = 0.0001) [20]. This concept is supported by another study which reported patients with complete resection of resistant disease (*n* = 7) showed significantly longer median time to progression than those with incomplete resections (*n* = 9; *p* = 0.014) [21]. The latter study also demonstrated that a small volume of resistant metastatic disease was associated with better clinical outcome after metastectomy [21]. All patients in these two studies received imatinib prior and post-surgery [20,21]. Another important factor to select appropriate surgical candidates in metastatic GIST is responding to systemic treatment and SD/limited PD prior to surgery which is supported by several studies. For example, one study reported post-surgery PFS and disease specific survival was 96% and 100%, respectively versus 0% and 60%, respectively at 12 months for patients responding to imatinib versus those progressing on imatinib [22]. Similar findings were observed in other studies [23,24]. Furthermore, patients on sunitinib seemed to have worse outcomes compared to patients on imatinib after metastectomy [25]. The PFS from metastectomy was 16 months on imatinib in contrast to 7 months of sunitinib [25]. These published results are consistent with our data, showing that the majority of patients who underwent metastectomy were on imatinib, had a small volume of metastatic disease and limited progression prior to surgery, had relatively good clinical outcomes of either NED, durable SD or PFS for more than 1 year, whereas a minority of patients who had shorter PFS, had significant residual metastatic disease after surgery. A few other studies looked specifically at liver metastectomy in combination with “adjuvant” imatinib showed improved OS compared to imatinib alone [26]. Almost all our patients continued imatinib after metastectomy and/or primary tumor removal with only one patient who did not tolerate the treatment, which may explain the relative prolonged mPFS (20.5 months) overall of metastatic patients who underwent surgery. It is interesting to note that mPFS of 20.63 months (95% CI: 14.52-not reached months) for de novo metastatic patients was almost twice as long as mPFS of 12.42 months (95% CI: 3.94-not reached months) for recurrent metastatic patients (Figure 1B,C). It is our belief, this difference is likely due to patient selection rather than disease biology.

The mOS of the entire metastatic GIST cohort (*n* = 127) was 48 (95% CI: 40.7 to 74.1) months with a 5-year survival of 42% (95% CI: 34% to 53%), which was previously reported in our recently published study [27]. We observed a significantly prolonged mOS of more than 8 years and a 5-year survival of 72% (95% CI: 0.58–0.89) (data not shown) in metastatic patients who underwent surgery. This is in contrast to a mOS of less than 4 years and a 5-year survival of 36% (95% CI: 25% to 50%) (data not shown) for metastatic cohort patients who did not undergo surgery. Although this difference is remarkable and highly statistically significant, it is important to recognize that patients who underwent surgery in the metastatic setting are highly selected and generally had favorable disease biology reflected by relatively good control of systemic treatment and a low volume of progressive disease, and therefore a favorable prognosis. Regardless, our finding is consistent with previously reported studies [20,21,22,23,24,25,26].

### 4.2. Radiotherapy (RT)

Current GIST international guidelines do not discuss the role of RT in these patients except for those with bone metastases [28]. This may be due to the perception of GIST as a radio-resistant tumor. In addition, it is often challenging to offer therapeutical meaningful but tolerable dose of RT given the GIST location and pattern of spread intra-abdominally. However, recent studies have highlighted the potential role of local control of RT as a treatment modality in the metastatic setting not only to palliate symptomatic local disease progression but also to eradicate resistant clones to prolong survival, similar to surgery as discussed above. This is likely due to the evolving improvement of RT techniques such as organ motion reduction, image guidance strategies and intensity-modulated radiation therapy (IMRT) [29,30]. In the largest prospective study to date of 25 patients, the majority of whom were on TKI, palliative RT was studied as a therapeutic option to target soft tissue metastasis (liver as well as other intra-abdominal metastasis) with results showing 80% achieved SD with a median duration of 16 months, while a minority of 8% achieved partial response, leading to the conclusion that RT may frequently result in durable stabilization for soft tissue metastasis therefore providing benefit for metastatic GIST patients by prolonging TKI treatment [31]. Another study similarly looked at the efficacy of RT in 15 locally advanced and metastatic GIST patients who were on TKI, and reported an estimated 6-month local PFS of 57% with no severe Grade ≥3 toxicities, and therefore concluded that RT can provide benefits in patients with metastatic disease without substantial additional toxicity [32]. The majority of patients in our cohort received palliative RT for symptomatic control, while only two patients received RT for local control with one patient achieving SD for 6 months whilst the other did not. Therefore, it is difficult to draw conclusions with regard to the benefit of RT to prolong PFS from our study and RT is generally not considered to be the “standard of care” in the absence of prospective trials. However, almost all patients in our cohort achieved good symptom control, supporting consideration of RT for both symptomatic bone and visceral metastasis in the palliative situation. Decisions for RT in the metastatic setting should be individualized and discussed in a multidisplinary context, taking into account the clinical situation, patient preference as well as the physician’s experience.

In terms of RT dose, 30 Gy at 3 Gy per fraction was commonly used with palliative intent [32]. However, higher dose RT (>5 Gy per fraction) has also been reported to provide a high response rate [32]. RT was administered concurrently with TKI in 41% of cases [32]. A total of 50 Gy was used concurrently with TKI in several case studies which showed favorable local disease control [33,34,35]. All these studies demonstrated not only efficacy but also safety with concurrent use of TKIs and RT which is reassuring and addresses an important practical question frequently raised in clinical practice. The majority of patients in our cohorts received between 20–40 Gy if the intent of RT was for palliation of symptoms, although a few patients received 50 Gy. In our cohort, all patients continued TKIs (mostly imatinib) during their RT treatment.

### 4.3. Local Ablation

RFA and MWA are minimally invasive procedures that induce coagulative necrosis of tumors using thermal energy. The role of RFA/MWA was first established in primary liver malignancy not amenable to surgery; however, it has also been used in other malignancies such as GIST patients with hepatic metastases. The evidence was scant with a few studies of limited sample sizes. One study from the US of nine patients with limited progressing metastatic liver disease despite continuing on imatinib and undergoing RFA with a short median follow-up of 4.2 months found five patients had PD within 6 months, and four patients remained SD (median follow up of 5.8 months) [36]. Another two international studies (one from Japan with seven patients and the other from Korea with 16 patients respectively) with long median follow ups of 31 and 33 months, reported a 4.2% and 6% local PD within 1 year, respectively. Both studies reported 100% GIST related OS at 5 years, albeit with a wide confidence interval due to a small sample size [37,38]. All these studies seem to demonstrate the safety and efficacy of RFA in treating limited progressive liver metastasis for GIST patients continuing on TKI treatments, which is consistent with our findings, albeit the sample size of our study was limited.

Other local ablative therapeutic approaches such as transarterial chemoembolization (TACE) and transarterial radioembolization (TARE) which involves the selective catheterization and delivery of microspheres coated with chemotherapy or yttrium-90 high dose radiation to target lesions, respectively, while sparing the normal hepatic parenchyma. Both of these locoregional therapies have been explored in metastatic GIST with some efficacy as well [39,40,41,42,43]. No patients underwent these locoregional treatments in our cohort.

### 4.4. Study Limitations

Due to the retrospective study design, our study could not collect the quality-of-life data which is important to address if LRT can improve the overall care of metastatic GIST patients. Another limitation of our study is the relatively limited sample size, in addition to it being retrospective in nature which may result in inherent bias. Propensity score matching analysis was attempted but unfortunately could not be performed because the matching cohort with similar disease and patient characteristics who did not undergo LRT could not be identified using our cohort.

## 5. Conclusions

Our study is among the first to systematically examine the practice pattern and clinical outcome of utilizing various LRT including surgery, RT and local ablation treatments such as RFA and MWA in metastatic GIST patients. The integration of these LRT into systemic treatments for carefully selected patients with a low volume of progressive metastatic GIST could be beneficial in prolonging their survival. Studies combining data from multiple institutions with larger sample sizes are warranted to further elucidate the value of LRT in the metastatic setting, and will be undertaken through the CanSaRCC national database (www.cansarcc.ca (accessed on 6 March 2022)) in the near future.

## Figures and Tables

**Figure 1 cancers-14-01477-f001:**
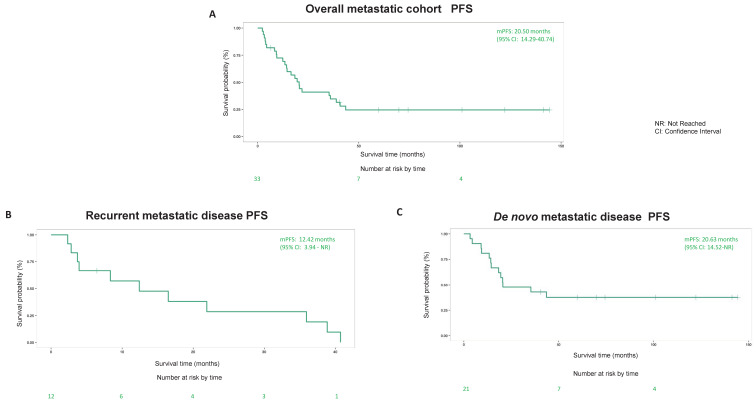
Clinical outcomes (PFS) of metastatic gastrointestinal stromal tumor (GIST) patients who underwent surgery. (**A**) PFS of overall metastatic cohort. (**B**) PFS of recurrent metastatic disease subgroup. (**C**) PFS of de novo metastatic disease subgroup.

**Figure 2 cancers-14-01477-f002:**
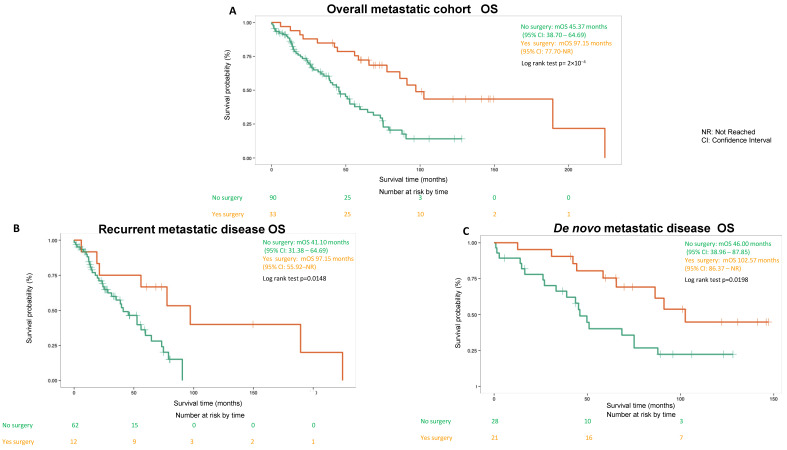
Clinical outcomes (OS) comparing metastatic gastrointestinal stromal tumor (GIST) patients who underwent surgery versus no surgery. (**A**) OS of overall metastatic cohort. (**B**) OS of recurrent metastatic disease subgroup. (**C**) OS of de novo metastatic disease subgroup.

**Figure 3 cancers-14-01477-f003:**
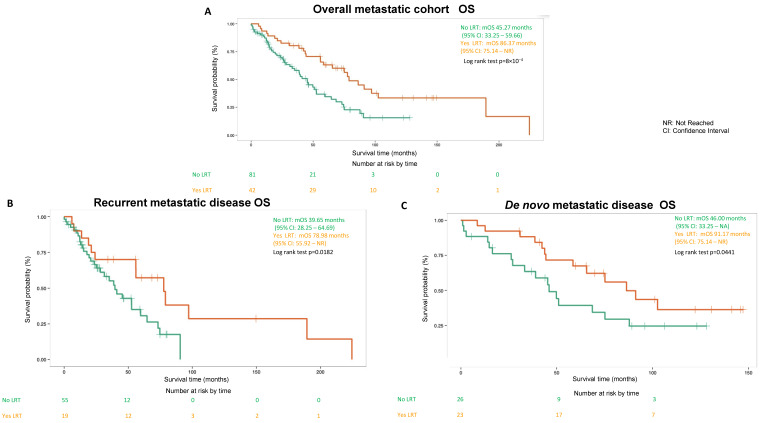
Clinical outcomes (OS) comparing metastatic gastrointestinal stromal tumor (GIST) patients who underwent LRT versus no LRT. (**A**) OS of overall metastatic cohort. (**B**) OS of recurrent metastatic disease subgroup. (**C**) OS of de novo metastatic disease subgroup.

**Table 1 cancers-14-01477-t001:** Overview of cohort characteristics–metastatic Gastrointestinal Stromal Tumors (GIST) in British Columbia (BC) (January 2008–December 2017).

	Patient Characteristics	Rec Met w/o LRT (%) *n* = 55	Rec Met with LRT (%) *n* = 20	De Novo Met w/o LRT (%) *n* = 26	De Novo Met with LRT (%) *n* = 26
	Age Median (min–max)	66 (31–84)	66 (31–75)	67.5 (23–90)	63 (36–82)
	Gender Female Male	21 (38.2%) 34 (61.8%)	9 (45%) 11 (55%)	15 (57.7%) 11 (42.3%)	8 (30.8%) 18 (69.2%)
	Treatment Center Vancouver Fraser Valley Victoria Interior Northern Community/unknown	18 (32.7%) 18 (32.7%) 10 (18.2%) 7 (12.7%) 1 (1.8%) 1 (1.8%)	12 (60%) 2 (10%) 1 (5%) 4 (20%) 1 (5%) 0 (0%)	17 (65.4%) 4 (15.4%) 2 (7.7%) 3 (11.5%) 0 (0%) 0 (0%)	7 (26.9%) 9 (34.6%) 4 (15.4%) 5 (19.2%) 1 (3.8%) 0 (0%)
**Primary Tumour Characteristics**	Tumour Size (cm) >2 ≥2 and ≤5 >5 and ≤10 10	1 (1.8%) 9 (16.4%) 16 (29.1%) 29 (52.7%)	0 (0%) 4 (20%) 8 (40%) 8 (40%)	1 (3.8%) 2 (7.7%) 8 (30.8%) 15 (57.7%)	0 (0%) 5 (19.2%) 14 (53.8%) 7 (26.9%)
	Mitotic Rate (/50HPF) <5 ≥5 and ≤10 >10 Unreported/unknown	16 (29.1%) 11 (20%) 23 (41.8%) 5 (9.1%)	3 (15%) 7 (35%) 10 (50%) 0 (0%)	10 (38.5%) 3 (11.5%) 5 (19.2%) 8 (30.8%)	7 (26.9%) 3 (11.5%) 12 (46.2%) 4 (15.4%)
	Tumour Location Stomach Small Bowel Rectum/Pelvis Other	23 (41.8%) 26 (47.3%) 3 (5.5%) 3 (5.6%)	5 (25%) 6 (30%) 6 (30%) 3 (15%)	7 (26.9%) 4 (15.4%) 4 (15.4%) 11 (42.3%)	7 (26.9%) 15 (57.7%) 1 (3.8%) 3 (11.6%)
	Tumour Rupture No Yes	49 (89.1%) 6 (10.9%)	19 (95%) 1 (5%)	26 (100%) 0 (0%)	22 (84.6%) 4 (15.4%)
	Mutational Status KIT Exon 11 KIT Exon 9 Wt PDGFRA Exon 18 D842V PDGFRA Exon 12 KIT Exon 9 and Exon 11 KIT Exon 11 and Exon 13 KIT Exon 11 and SDHA Unknown/failed	*n* = 41 (ordered) 20 (48.8%) 7 (17.1%) 7 (17.1%) 2 (4.9%) 1 (2.4%) 1 (2.4%) 1 (2.4%) 0 (0%) 2 (4.9%)	*n* = 16 (ordered) 11 (68.7%) 2 (12.5%) 3 (18.8%) 0 (0%) 0 (0%) 0 (0%) 0 (0%) 0 (0%) 0 (0%)	*n* = 15 (ordered) 8 (53.3%) 0 (0%) 3 (20.0%) 0 (0%) 0 (0%) 0 (0%) 0 (0%) 1 (6.7%) 3 (20.0%)	*n* = 17 (ordered) 13 (76.4%) 1 (5.9%) 2 (11.8%) 0 (0%) 0 (0%) 0 (0%) 0 (0%) 0 (0%) 1 (5.9%)
	Systemic treatment median (range)	1 (1–4)	2 (1–5)	2 (1–5)	1 (1–4)

**Table 2 cancers-14-01477-t002:** Clinical characteristics and outcome of metastatic gastrointestinal stromal tumor (GIST) patients who underwent radiotherapy (RT).

Patient Characteristics				Primary Tumour Characteristics				Treatment Characteristics and Outcome		
Patient ID	Age at Metastatic Diagnosis	Gender	Recurrence/De Novo	Size (cm)	Location	Mitotic Count (/50HPF)	Tumor Rupture (Yes/No)	Systemic Treatment *	Reason for RT	Clinical Outcome
22	79	Male	Recurrence	10.5	Stomach	5	No	I, S ^1^	Bleeding	SI
27	59	Female	Recurrence	7	Rectum	>10	No	I, S, R, C ^1^, Ri,	Pain	NSI
38	71	Female	de novo	5	Stomach	10–15	No	I ^1^, S	Bleeding	SI
89	63	Male	de novo	3.7	Stomach	150	No	I, S ^1^	Preventative bleeding **	SI
97	67	Male	Recurrence	6.3	Rectum	20	No	I ¹	Local control for progressive bone disease	SD for 6 months
99	57	Male	Recurrence	16.9	Rectum	100	No	I (Adjuvant) I ^1^, R, S	Pseudo-adjuvant RT after metastectomy	Local control not achieved
106	72	Female	de novo	9.2	Small bowel	Not reported	No	I, S ^1^	Pain	SI
143	73	Female	Recurrence	5.1	Large bowel	3–4	No	I (Adjuvant) I ^1^, S	Pain	SI
149	78	Female	Recurrence	8.5	Rectum	50	No	I (Neoadjuvant) S ^1^	Pain	SI
269	76	Male	Recurrence	14	Stomach	10	No	I (Adjuvant) I ^1^, S, R, I, A	Bleeding	SI
358	63	Male	Recurrence	18	Small bowel	100	No	I (Neoadjuvant) I ^1^, R, A ^1^	Pain	SI
414	36	Male	Recurrence	5.5	Small bowel	30	No	I, S, R, N,	Local control for progressive visceral disease	Local control not achieved

RT—radiation therapy; I—imatinib; S—sunitinib; R—regorafenib; C—cabozantinib; Ri—ripretinib; A—avapritinib; N—nilotinib; So—sorafenib; SI—symptom improvement; NSI—no symptom improvement; SD-stable disease. * if not specified (neoadjuvant/adjuvant), systemic treatment is for metastatic disease which is listed in sequence. ** Potential bleeding of tumor due to patient being given anticoagulation medication for treatment of splenic vein thrombus. ^1^ systemic treatment that patient was on during RT.

**Table 3 cancers-14-01477-t003:** Clinical characteristics and outcome of metastatic gastrointestinal stromal tumor (GIST) patients who underwent local ablation.

Patient Characteristics				Primary Tumour Characteristics				Treatment Characteristics			
Patient ID	Age at Metastatic Diagnosis	Gender	Reccurence/De Novo	Size (cm)	Location	Mitotic Count (/50HPF)	Tumour rupture (Yes/No)	Systemic Treatment *	Reason for Ablation	Best Response	Duration of Response
185	80	Female	Recurrence	4	Small bowel	<5	No	I ¹	MWA–2 liver lesions	PR	17 mos
214	48	Male	de novo	9.5	Small bowel	<5	No	I (Neoadjuvant) I ^1^, S, R, I	RFA–1 liver lesion	PR	12 mos
413	57	Female	Recurrence	7	Stomach	12–16	No	I ^1^, S, So, I	RFA–2 liver lesions	PR	1 mos

MWA—microwave ablation; RFA—radiofrequency ablation; PR—partial response; I—imatinib; S—sunitinib; R—regorafenib; So—sorafenib; mos—months. * if not specified (neoadjuvant/adjuvant), systemic treatment is for metastatic disease which is listed in sequence. ^1^ systemic treatment that patient was on during ablation.

## Data Availability

Not applicable.

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
