# Peer review of "Locoregional Treatments for Metastatic Gastrointestinal Stromal Tumor in British Columbia: A Retrospective Cohort Study from January 2008 to December 2017"

_cancers, 2022, doi:10.3390/cancers14061477_

Round 1
Reviewer 1 Report
This paper is well-written, off course there were still biases caused by retrospective cohort. But the readers of Cancers are interested in this paper. I hope prospective trial or propensity matching analysis in next research.
This manuscript is a resubmission of an earlier submission. The following is a list of the peer review reports and author responses from that submission.
Round 1
Reviewer 1 Report
The idea for the article "Locoregional Treatments for Metastatic Gastrointestinal Stromal Tumor in British Columbia: a retrospective cohort study from January 2008 to December 2017" has been conceived and executed in a meticulous way. The authors have systematically analyzed the outcome of a decade long study which will be the addition of experimental evidence for the treatment of metastatic GIST patients with low volume disease.
This paper can be accepted in its original form for publication
Author Response
Thanks so much for the comment and support
Reviewer 2 Report
The authors reported the paper of “Locoregional Treatments for Metastatic Gastrointestinal Stromal Tumor in British Columbia: a retrospective cohort study from January 2008 to December 2017” and concluded that integration of LRT with systemic treatments may potentially provide promising durable response and prolonged survival for highly selected metastatic GIST patients with low volume disease, limited progression and otherwise well controlled on systemic treatments.
This paper is well-written, but I think there are some problems in this paper.
- The selection of standard therapy should be established by prospective clinical trial. The results of this study has big biases from the retrospective cohort.
- At least, propensity score matching analysis or multivariable analysis should have been performed to decrease biases from retrospective cohort.
- The US, Europe, or Japanese guidelines for GIST did not define RT as a standard therapy for metastatic GIST. The authors should describe why he (patient A) selected RT therapy, or she (patient B) did not.
Author Response
Dear reviewer
We greatly appreciate your time and efforts to provide valuable suggestions and comments on our manuscript. We have taken these suggestions/comments seriously and modified our manuscript accordingly. Please see attached revised manuscript with track changes. We have also addressed these comments on a point-by-point basis as per below.
Reviewer 2
The authors reported the paper of “Locoregional Treatments for Metastatic Gastrointestinal Stromal Tumor in British Columbia: a retrospective cohort study from January 2008 to December 2017” and concluded that integration of LRT with systemic treatments may potentially provide promising durable response and prolonged survival for highly selected metastatic GIST patients with low volume disease, limited progression and otherwise well controlled on systemic treatments.
This paper is well-written, but I think there are some problems in this paper.
- The selection of standard therapy should be established by prospective clinical trial. The results of this study has big biases from the retrospective cohort.
Response: Thanks for the reviewer’s comment. We agree with the reviewer that all retrospective studies, have inherent biases since the study operations, data collection, data entry, data quality assurance, and so on are not preplanned and controlled in the same manner as gold standard prospective clinical trial. Therefore, the conclusion from retrospective studies are not definitive, and should not be considered as standard of care unless it is further validated by prospective clinical trials. Therefore throughout our manuscript, we have carefully chosen appropriate words in discussing the results and making reasonable conclusions (eg: LRT may be considered in selective patients with specific clinical characteristics). We did not conclude that this should be a “standard of care”. Based on the fact that the finding from our study is consistent with all previous published retrospective studies (which are all limited by bias), we believe that our conclusion is reasonably sound. It would be ideal to confirm our findings and others in a large randomized prospective clinical trials, but this was attempted a few times in different countries and unfortunately failed due to low accrual. This is not surprising given GIST is a rare cancer with an incidence of 6 in a million per year as well as logistic reasons (eg. Patient reluctance to undergo surgery or not to undergo surgical). We have discussed this in our manuscript (Line 251-284).
- At least, propensity score matching analysis or multivariable analysis should have been performed to decrease biases from retrospective cohort.
Response: Thanks for the reviewer’s great suggestion. We agree with the reviewer that propensity score matching analysis is a common analysis to potentially remove confounding bias in nonrandomized cohort study. We have attempted this analysis but were unsuccessful, likely limited by relative small sample size (Total N=127 patients with N=52 de novo metastatic disease and N=75 recurrent metastatic disease). It is important to note in our study that the group who underwent LRT are a highly selected group of patients with small volume of disease, oligoprogressive disease, younger age and good performance status, compared to the group who did not undergo LRT. Therefore, it would be very challenging to identify a matching cohort with similar disease characteristics (small volume of disease and oligoprogressive disease) and patient characteristics (age and performance status) that did not undergo LRT to do the propensity score matching analysis. We have added this limitation in our discussion (Line 354-357).
However, we have performed multivariate cox regressions to evaluate the associate with surgery/LRT and OS, adjusted by de novo/recurrent met status, age, sex, tumor size and mitotic count. The conclusion remains the same. For surgery vs non-surgery group comparison, the hazard ratio (HR) of OS is 0.343 (p=0.0004) without adjustment, and the HR of OS is 0.393 (p=0.004) after the multivariate regression adjustment. For LRT vs non-LRT group comparison, the hazard ratio (HR) of OS is 0.416 (p=0.0011) without adjustment, and the HR of OS is 0.460 (p=0.01) after the multivariate regression adjustment. We have added this method and result in in our revised manuscript. (Line 107-110; 235-241)
- The US, Europe, or Japanese guidelines for GIST did not define RT as a standard therapy for metastatic GIST. The authors should describe why he (patient A) selected RT therapy, or she (patient B) did not.
Response: We agree with the reviewer that RT is not considered a “standard treatment” for localized and metastatic GIST as GIST anecdotally is considered as “radioresistant”. We reported the palliative benefit of RT (ie; bleeding and pain control) as well as occasional good local control of RT in metastatic GIST, which is consistent with other published reports (discussed in our manuscript). The selection of RT for a particular patient with metastatic GIST should remain individualized within a multidisciplinary context, taking into account the clinical situation, patient preference as well as the physician’s experience. We have added this discussion in our revised manuscript (Line 329-331)